# A Review of Searches for Evidence of Tachyons

Robert Ehrlich

Physics and Astronomy Department, George Mason University, Fairfax, VA 22030, USA; rehrlich@gmu.edu

**Abstract:** Here, we review searches for empirical evidence for the existence of tachyons, superluminal particles having $m^2 < 0$. The review considers searches for new particles that might be tachyons, as well as evidence that neutrinos are tachyons from data that may have been gathered for other purposes. Much of the second half of the paper is devoted to the $3 + 3$ neutrino model including a tachyonic mass state, which has empirical support from a variety of areas. Although this is primarily a review article, it contains several newly identified results.

**Keywords:** neutrinos; tachyons; $3 + 3$ model





## 1. Introduction

This review of searches for evidence of tachyons follows the only other one ever performed a half century ago by Michael Kreisler [1]. Like that first review, this one can make no claim that "extraordinary" evidence (Sagan's Criterion) [2] for tachyons has been found, but unlike Kreisler's review, this one points to many pieces of circumstantial evidence that tachyons actually do exist, or more specifically, that some neutrinos are tachyons. The evidence is from a wide range of fields including cosmic rays, supernovae, cosmology, and neutrino lab experiments. Possible problems with the tachyonic neutrino hypothesis are also discussed, but none of them appear to be fatal. In fact, some of the often-expressed concerns about tachyons, e.g., their lack of respect for causality, or the reversals of cause and effect they might permit, could be viewed as an argument for their existence, rather than the reverse, because that is the case in quantum mechanics, which operates equally well in forward or reverse time. Furthermore, as Andrzej Dragan and Artur Ekert have shown [3], if one keeps both subluminal and superluminal terms in the Lorentz Transformation, then the non-deterministic behavior and non-classical motion of particles arise as a natural consequence, similar to quantum theory

## 2. Tachyons as New Particles

In 1962, Bilaniuk, Deshpande, and Sudarshan [4] suggested a way superluminal particles, now known as tachyons, might exist without conflicting with Einstein's ban on faster-than-light speeds provided they had such speeds from the moment of their creation. Einstein himself, in fact, cautiously restricted his ban on superluminal speeds to the case of particles that were slowly accelerated and had an initial speed less than light. In fact, he noted in his 1905 paper that: "For velocities greater than that of light our deliberations become meaningless" [5]. After the work by Sudarshan and colleagues in 1962, it was Gerald Feinberg who, later, in a classic paper, coined the term tachyon and proposed that tachyonic particles could be made from excitations of a quantum field with imaginary mass [6].

After 1962, physicists searched for new particles whose properties matched those of tachyons, specifically they either traveled at superluminal speed or had a negative mass squared. The former category would presumably emit Cherenkov radiation in a vacuum if they were charged. In several experiments it was assumed that tachyons of charge $Ze$ could be produced in lead by 1.2 MeV photons from a $^{60}Co$ source, after which they passed between a pair of charged plates having an electric field $E$ between them [7,8]. It was

assumed that the rate of change in their kinetic energy with distance due to the Cherenkov effect (first term) and the electric field (second term) is given by:

$$\frac{dK}{dx} = \frac{-2\pi^2 K^2 Z^2}{h^2 c^2} + ZeE \tag{1}$$

For a significant plate separation, the tachyon would quickly attain a kinetic energy $K$ at which $dK/dx = 0$. Thus, experimenters looked for a peak above background at that predicted energy when particles were detected. In the first experiment, one detector was used, while the second experiment used two detectors in coincidence to reduce the background, but neither experiment found any evidence for tachyons. The reported negative results of these experiments meant that if tachyons were being produced, their cross-section was more than $10^8$-times smaller than that of electron–positron pair production at the same 1.2 MeV energy.

It is possible that tachyons might only be produced at very high energy. In particular, they might be present in the showers created when extremely high-energy cosmic rays enter the Earth's atmosphere from space and could reveal their presence if they reached ground-based detectors before the shower front traveling at essentially light speed. Here again, only upper limits were found, although an initial 1974 report by Roger Clay and Philip Crouch reported seeing "possible" evidence for tachyons in the form of many pulses in the detectors in advance of the shower front [9]. The same group, however, using an improved electronics system, failed to reproduce their earlier results, which was also the case in subsequent searches by other groups conducted during the 1970s and early 1980s [10–17]. A much more recent search in 2020 similarly found no evidence in cosmic ray data [18].

Regarding the lab experiments looking for tachyons from non-astrophysical sources, one method was based on the observed "missing mass." For example, consider the reaction: $K^- + p \rightarrow \Lambda^0 + X^0$, where $X^0$ is a neutral particle. Even though this neutral particle would leave no tracks in a bubble chamber assuming it is long-lived, the neutral $\Lambda^0$ does reveal the particle's momentum vector when it decays via $\Lambda^0 \rightarrow p + \pi^-$, and so, one could easily compute the unseen $X^0$ mass based on track measurements for all charged particles and then imposing energy and momentum conservation. Figure 1 shows the results of one such missing mass experiment by Baltay et al. [19]. The narrow peak at $m_X^2 = 0.02$ GeV$^2$ corresponds to X being a neutral pion. Most of the rest of the events having $m_X^2 > 0$ have two neutral pions. In principle, the small number of events having $m_X^2 < 0$ are tachyon candidates, but when the tracks for these events were remeasured, virtually all of them turned out to have $m_X^2 > 0$, so they just represent bad measurements. A year after Baltay et al., Danburg et al. [20] obtained the same negative result from the same reaction.

Another bubble chamber experiment looked for tachyons being emitted when no incident particles were present, but where protons in the chamber underwent decay via the reaction: $p \rightarrow p + T^0$ and $p \rightarrow p + T^0 + \bar{T}^0$. These processes could only occur if the tachyon had a negative energy, which is allowed for these particles, if they exist. The negative result in this case was that if the process does occur, its lifetime would be at least $2 \times 10^{21}$ years [21]. Further limits on the process were later set by Ljubičic et al. using a somewhat different technique [22].

A completely different type of search involved looking for tachyon-like magnetic monopoles. The motivation for this odd-sounding possibility was based on the fact that ordinary particles are either neutral or electrically charged, while those that move at $v = c$ are neutral. Therefore, symmetry might require tachyons to be either neutral or magnetically charged. Based on this conjecture, Bartlett and colleagues conducted several searches for tachyon magnetic monopoles (TMs) having $v > c$ [23,24]. These searches looked for particles emitted from a radioactive source and later cosmic ray particles that emitted Cherenkov radiation in a vacuum. In principle, the TM gains energy in a magnetic field (just like an electric charge gains energy in an electric field), and it would rapidly attain an equilibrium energy where the energy gained equals the energy lost due to Cherenkov

radiation, which is then detected by photomultiplier tubes. Given all the theoretical unknowns concerning tachyon monopoles, Bartlett et al. conclude their 1972 article by listing four reasons why they might not have detected them even if they exist. While Bartlett et al. reported negative results for searches for TMs from both of their searches, some "possible" positive results were reported by Fredericks [25]. Those results, however, relied on an entirely different, and somewhat subjective, hard-to-quantify technique (the exact shape of particle tracks in a nuclear emulsion).

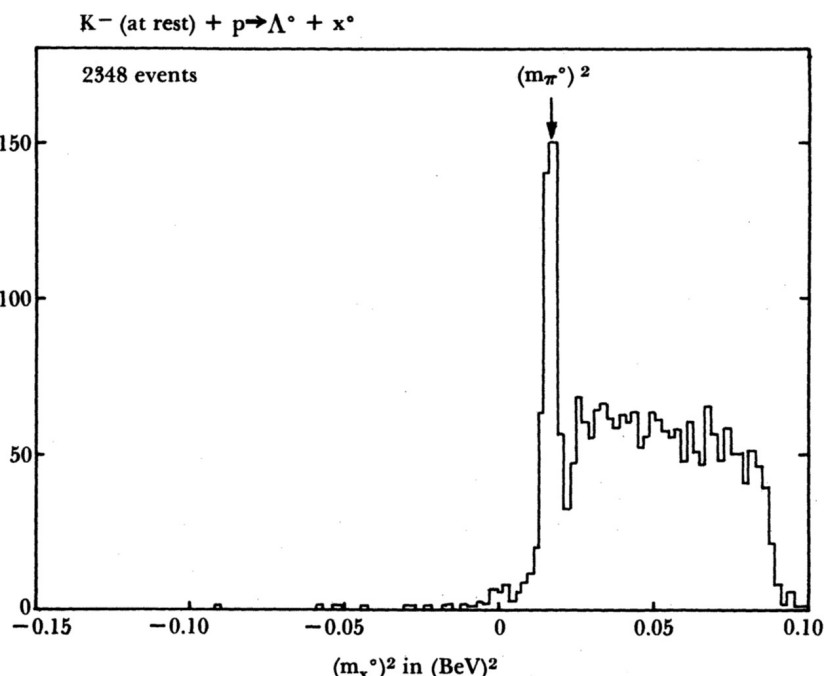

**Figure 1.** Plot of missing mass, that is the mass of X for the reaction $K^- + p \to \Lambda^0 + X$.

A most unusual experiment searching for tachyons was performed by V. P. Perepelitsa in 1977 [26]. This experiment, dubbed a "tachyon Michelson experiment", looked for the possible tachyon reaction $e^+ e^- \to T^+ + T^-$ using positrons at rest from a radioactive source. The basis of the name of the experiment was that Perepelitsa believed that in order to prevent a causality violation with tachyons, their emission directions would be restricted to certain angles between the velocity of the particle and the instantaneous velocity of the laboratory with respect to some preferred reference frame. Under this assumption, tachyons that corresponded to events in two counters that corresponded to $v >> c$ times-of-flight would have a distribution in time that varies due to the Earth's rotation, and hence with celestial time. No such variation was seen, and Perepelitsa was able to set an upper limit on the branching ratio of tachyon production relative to photons. Of course, this negative result depended on his assumption that tachyons, if they exist, could not violate causality, which is a feature of some theories of these particles that stipulate the existence of a preferred reference frame [27,28].

One final search noted here for tachyons as new particles was also performed by Perepelitsa and colleagues in 2019 and 2020. The group reported positive results in a pair of unpublished papers [29,30]. Perpelitsa and colleagues searched for anomalous Cherenkov rings indicating superluminal speed charged particles as recorded by the DELPHI detector at CERN from high-energy $e^+ + e^-$ collisions. However, the authors note in their abstract that "the opinions, findings and conclusions expressed in this material are those of the authors alone and do not reflect in any way the views of the DELPHI Collaboration." In their experiment, they found 27 candidates for tachyons based on the Cherenkov ring radii and the measured track momenta, which allowed them to infer $m^2 < 0$ values. Examining the distribution of the magnitude of the tachyon mass ($m = \sqrt{-m^2}$), they found two peaks

corresponding to $m_1 = 0.29 \pm 0.01$ GeV/c$^2$ and $m_2 = 4.6 \pm 0.2$ GeV/c$^2$. Given the low statistical significance of the claim ($p < 10^{-3}$), however, more data are clearly warranted before this unpublished claim can be considered to be credible.

The long string of negative, low statistics, and ambiguous results from searches for tachyons may have suggested to many physicists that tachyons do not exist. However, such an inference would be unwarranted. Perhaps one was looking for them in the wrong way or in the wrong places, or perhaps, their cross-section was simply very small. Even more simply, perhaps they are actually a well-known particle, rather than a new one.

## 3. Neutrinos as Tachyons

In 1970, Cawley was the first to raise the possibility that neutrinos might be tachyons [31], but it was not until 1985 that Chodos, Hauser, and Kostelecký [32] discussed a theoretical framework for describing tachyonic neutrinos. While Chodos and collaborators were candid about the difficulties of formulating a complete quantum field theory of tachyonic neutrinos, they also noted that such difficulties cannot be used to exclude a priori their existence. Indeed, since 1985, theoreticians including Jentschura [33], Rembieliński and Ciborowski [34], Radzikowski [28], and Schwartz [35] have found ways to deal with many of these difficulties.

Chodos et al.'s suggestion that neutrinos might be tachyons was a natural one because, unlike all other particles, no neutrinos had then (or now) ever been observed to travel slower than light or have a positive mass squared. In fact, of the 13 experiments to measure the effective mass of neutrinos between 1990 and 2021, all but one reported $m^2 < 0$ values, two of the values being many standard deviations below zero [36]. None of the researchers reporting those negative $m^2$ results, however, regarded them as constituting evidence for electron neutrinos being tachyons, and indeed, the experimenters sometimes described their results to be "unphysical", being due to some unknown or suspected systematic effect.

In recent years, the search for evidence of tachyonic neutrinos has been very well motivated by the increasing recognition that Lorentz invariance may be slightly violated, and the neutrino sector is probably the most likely sector where such a violation might be observed [37]. Superluminal neutrinos would not be the only way to have a violation, but it is certainly one way it could occur, so theorists open to the possibility of small violations of Lorentz invariance should also be open to the possibility of tachyonic neutrinos. Alan Kostelecky and colleagues have been particularly active in formulating the Standard-Model Extension (SME), which is an effective field theory containing all operators for different types of Lorentz violation. They have also been very active in proposing tests to look for each type of violation [38].

Before considering experiments on the neutrino mass to look for possible $m^2 < 0$ values, let us first consider measurements of neutrino speed. One could, for example, demonstrate that neutrinos are superluminal if they traveled a precisely determined distance in less time than light in a vacuum. This time-of-flight technique cannot be performed using individual neutrinos, given their low cross-section, but rather with short bunches of neutrinos produced in a pulsed accelerator, reactors, of course, being unable to produce short pulses of neutrinos. The most memorable such neutrino velocity measurement was made by the OPERA Group at CERN, which reported in 2011 that bunches of muon neutrinos in a 17 GeV beam had apparently traveled a distance of 732 km in a time that was $60.7 \pm 9.9$ ns less than a photon, a $6\sigma$ result. The corresponding excess above light speed was $\delta = v/c - 1 = 2.48 \pm 0.42 \times 10^{-5}$ [39].

The OPERA initial result was the stimulus for an impressive number of theoretical papers, some of them very skeptical. In fact, one paper by Cohen and Glashow showed that given the large size of $\delta$ reported, the result would be the radiation of electrons and positrons through the process $\nu_\mu \rightarrow \nu_\mu + e^+ + e^-$, which could only occur if neutrinos are superluminal [40]. Cohen and Glashow never used the word tachyon or considered superluminal neutrinos to have $m^2 < 0$, but they showed that the energy threshold for the above decay process to occur was given by $E = 2m_e/\sqrt{\delta} \sim 140$ MeV, $m_e$ being the electron

mass. They further showed that for 17 GeV neutrinos (three orders of magnitude above the threshold), their resulting energy loss was great enough to make it impossible for most of the neutrinos in the beam to reach the detectors, meaning that the OPERA initial result had to be in error.

This OPERA neutrino anomaly was subsequently shown to be an error resulting from a loose cable and a clock that ticked too fast. The two errors had opposite impacts on the measured neutrino speed, but when they were both corrected, the excess above light speed was only $\delta = 2.7 \pm 5.1 \times 10^{-6}$ [41]. Similar measurements of the speed of muon neutrinos in pulsed accelerator beams at GeV-scale energies have been made by the ICARUS, MINOS, and T2K groups. All such speed measurements have resulted in values consistent with that of light within experimental uncertainties [42–44]. In summary, these speed measurements neither required nor rejected the possibility of superluminal neutrinos; they just were not sensitive enough to measure any nonzero value of $\delta$ given the smallness of the neutrino mass. For example, if we use the limit on $\delta = 10^{-6}$ from the MINOS Collaboration, [43] with their neutrino beam energy $E = 3$ GeV, we find $m \sim \sqrt{2E^2\delta} = 4$ MeV, which is more than an order of magnitude greater than the current upper limit on the muon neutrino mass ($m_\nu < 0.16$ MeV) [45].

## 4. Hints from IceCube Data

If tachyonic neutrinos satisfy the usual relation between velocity and energy, we would expect their superluminality to become most evident the lower their energy; in fact, zero-energy tachyonic neutrinos would have infinite speed. However, it is also possible that the reverse is true if a different dispersion relation, $v = v(E)$, were to apply for $v$ close to $c$, in which case, the degree of superluminality might increase with the neutrino energy. The experiment observing the highest-energy neutrinos probably of extragalactic origin is IceCube, which has observed neutrinos having $E_\nu > 60$ TeV.

If superluminal neutrinos are present, their spectrum might have a cut-off due to their creation of Cherenkov radiation when they undergo the vacuum pair emission (VPE) process $\nu \rightarrow \nu + e^+ + e^-$, originally suggested by Cohen and Glashow, a process that is only possible for $v > c$ particles [40]. From initial data, it appeared that the spectrum of IceCube neutrinos did have a cut-off above 2 PeV, which would correspond to $\delta \sim 0.75 \pm 0.25 \times 10^{-20}$ [46,47].

However, more recent data showed that there appeared to be no cut-off up to an energy of 10 PeV, the highest energy for which data are available, and in fact, the spectrum is consistent with a single power law, although more complex variations cannot be excluded [48]. In an effort to keep the superluminal interpretation alive, Jiajun Liao and Danny Marfatia suggested the IceCube data were consistent with a two-component spectrum of both superluminal and subluminal neutrinos and CPT violation [49].

While the Liao–Marfatia interpretation is open to question, a completely different claim of superluminal neutrinos in IceCube data was made by Yanqi Huang and Bo-Qiang Ma in 2018, who found nine instances in which extremely high-energy neutrinos arrived at about the same time and came from the same direction as a gamma-ray burst (GRB) observed by other instruments [50]. Five of the nine neutrinos appear to have traveled very slightly faster than c, while the other four traveled very slightly slower, and there was a trend of linearly increasing $\delta$ with increasing neutrino energy.

After a follow-up search published a year later, they found 12 more neutrinos associated with GRBs satisfying a similar pattern [51]. Huang and Ma suggested that this finding represents Lorentz violation of cosmic neutrinos and also a violation of CPT symmetry between neutrinos and antineutrinos. If CPT were violated, neutrinos and anti-neutrinos would, of course, have opposite signs for the Lorentz violating factor, so the neutrinos could be superluminal and antineutrinos subluminal, or the reverse. Note that as with Cohen and Glashow, Huang and Ma never refer to superluminal neutrinos as being tachyons having an imaginary mass, but prefer to frame their analysis in terms of a Lorentz violating dispersion relation.

However, Ellis et al. [52] argued that the significance of Huang and Ma's result is overstated because it depends on cross-correlating sub-MeV and multi-GeV photons from GRBs, which is risky given the paucity of multi-GeV photons from these occurrences. Casting further doubt on the Huang–Ma result, a search for low-energy ($E < 17.5$ MeV) neutrinos associated with GRBs (within a time window of $\pm 500$ s) in the KamLAND detector has yielded no candidates above background [53]. In summary, like the Liao–Marfatia analysis, that by Huang and Ma cannot be said to have yet provided definitive evidence for superluminal neutrinos or for violations of CPT or Lorentz invariance.

## 5. Energetically Forbidden $\beta$−Decay

Following their 1985 paper suggesting that some neutrinos might be tachyons, Chodos and his colleagues, Kostelecky, Potting, and Gates, considered in 1992 a most surprising experimental test of this possibility [54]. Consider, for example, the following two energetically forbidden examples of beta decay:

$$p \rightarrow n + e^+ + \nu_e \quad (-1.8\,\text{MeV}) \tag{2}$$

$$^4He \rightarrow^4 Li + e^- + \bar{\nu}_e \quad (-22.9\,\text{MeV}) \tag{3}$$

where the numbers in parenthesis indicate the deficit in energy preventing the decay from occurring. Chodos and colleagues realized that if the electron neutrino emitted in beta decay is a tachyon, then its total energy $E$ could be negative in some reference frames, and thereby permitting the two energetically forbidden decays if the energy of the tachyonic neutrino or antineutrino was sufficiently negative, e.g., more negative than $-1.8$ MeV for reaction 2. Tachyons, of course, just like photons, have no rest frame. Thus, it is not possible to perform a Lorentz Transformation (LT) from the lab frame to the tachyon rest frame. One can, however, perform an LT of the tachyon neutrino energy in going from the lab frame (E) to the rapidly moving proton rest frame (E′):

$$E' = \gamma[E - \beta p cos(\theta)] \tag{4}$$

If the angle of the emitted neutrino $\theta = 0$ and $E > 0$, we find that an observer having a speed $\beta > E/p = E/\sqrt{E^2 - m^2 c^4} < 1$ (since $m^2 < 0$) would judge the emitted neutrino energy $E'$ to be negative. A consequence of this fact is that any energetically forbidden beta decay becomes allowed when the parent nucleus moves past us at sufficient speed or energy. For example, from relativistic kinematics, the thresholds for the decays given in Equations (2) and (3) can be easily shown to be:

$$E_{p,th}(\text{eV}) = \frac{1.8 \times 10^{15} m_p(\text{GeV})}{|m_\nu|(\text{eV})} \tag{5}$$

$$E_{\alpha,th}(\text{eV}) = \frac{22.9 \times 10^{15} m_{He}(\text{GeV})}{|m_\nu|(\text{eV})} \tag{6}$$

Obviously, based on Equations (5) and (6), if the magnitude of the tachyonic neutrino mass $|m_\nu| \equiv \sqrt{-m_\nu^2}$ is very small, the threshold energy for proton beta decay (Equation (5)) will be very high, while that for alpha decay (Equation (6)) will be 50.9-times higher still. Chodos et al. suggested the beta decay: $^{162}Dy \rightarrow^{162} Ho + e^- + \bar{\nu}_e$ as being a particularly promising one to look for in view of its low threshold energy, about a third of that for proton decay [54]. However, this assessment ignores the difficulty of producing a very pure beam of high-intensity and high-energy $^{162}Dy$ nuclei compared to protons and the fact that the energy required might easily exceed that of any known accelerator, depending on the value of $|m_\nu|$.

## 6. The Cosmic Ray Spectrum

Cosmic rays consist of protons and other nuclei that bombard the Earth from space with an enormous range of energies, some far higher than any accelerator. The majority

of the cosmic rays are protons and alpha particles with small contributions from heavier nuclei. Their spectrum is approximately given by a power law in their flux of the form: $dF/dE \propto E^{-\gamma}$ over many decades of energy $E$, where $\gamma \approx 3$. There are four notable departures from a power law in the cosmic ray spectrum that have been the subject of great interest to researchers. These four features are known as the first and second "knees", the "ankle", and the GZK cut-off. The two knees are slight increases in the value of $\gamma$ (a "hardening" of the spectrum) that abruptly occur at specific energies, while the ankle is the reverse, an abrupt decrease in $\gamma$.

The GZK cut-off is a hypothesized abrupt end of the spectrum suggested independently by Greisen, Zatsepin, and Kuzmin [55,56] at an energy of about $5 \times 10^{19}$ eV. At that energy, cosmic ray protons would have enough energy to be blocked by their interaction with the cosmic background radiation filling all space. The reaction accounting for this cut-off would be $p + \gamma \rightarrow \Delta^*(1238) \rightarrow p + \pi$, where $\gamma$ is a background radiation photon and $\Delta^*(1238)$ is a well-known resonant state of a proton and pion. If the cosmic ray particle is a nucleus of atomic mass A instead of a proton, then the predicted cut-off would be A-times higher energy, because it is really the energy per nucleon in the nucleus that determines the threshold of the $\Delta^*$ reaction.

The presence of the two knees and ankle shows up most clearly when one displays $E^3 dF/dE$ rather than $dF/dE$ itself—see Figure 2. The existence of the GZK cut-off is not obvious in the figure, in light of the handful of data points above $E = 10^{20}$ eV, in seeming defiance of the cut-off. However, the rapid decline of $E^3 dF/dE$ for the more accurate (blue) data points between the dashed line and $E = 10^{20}$ clearly shows the cut-off occurs at the predicted energy (for a proton). The four features of the cosmic ray spectrum noted above have a natural explanation if the electron neutrino is a tachyon of mass $|m_\nu| \approx 0.5$ eV or $m_\nu^2 \approx -0.25$ eV$^2$, as I discussed in a pair of papers in 1999 [57,58]. However, it should be noted that this mass estimate was only an approximate value, given the uncertainty in the energy of the knees, and in a later paper, I revised it to $m_\nu^2 = -0.11$ eV$^2$ based on other evidence [59].

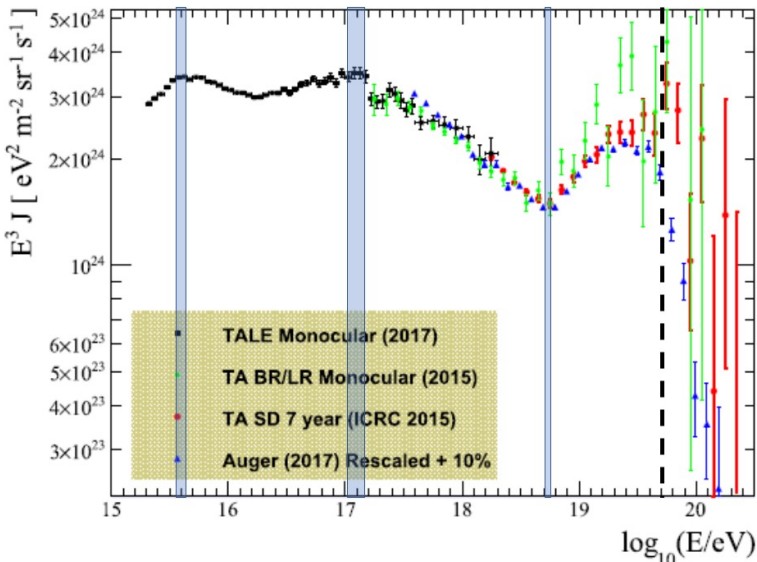

**Figure 2.** Plot of $E^3 J \equiv E^3 dF/dE$ for the cosmic ray spectrum using data from the four indicated experiments showing the locations of the two knees, the ankle, and the GZK cut-off.

The data are from [60], and the three vertical bands have been added to show the approximate uncertainty in the energies of the two knees (downturns) and the ankle (an upturn). The dashed line shows the energy of the hypothesized GZK cut-off.

Suppose one assumes a threshold energy for proton beta decay (Equation (2)) of 3 PeV = $3 \times 10^{15}$ eV (the energy of the first knee). In that case, at higher energies, protons

are depleted from the spectrum in increasing numbers, which explains the occurrence of the first knee, assuming the cosmic ray spectrum at their source is a pure power law, i.e., no knee. Based on Equation (5), we find for the mass of the tachyonic electron neutrino $|m_\nu| \approx 0.5$ eV, a fact that Alan Kostelecky had originally noted [61]. Cosmic ray researchers, of course, have a more conventional explanation of the knee, namely it is believed to be the transition from galactic to extragalactic cosmic rays. That standard interpretation, however, remains unverified since we cannot yet identify the locations of cosmic ray sources, and Erlykin and Wolfendale claimed the knee is too sharp to be explained in that manner [62]. Moreover, the tachyonic interpretation is buttressed by the observation that, while the fraction of the spectrum that consists of protons is rising before the knee, it starts declining right after that energy [63]. The existence of the second knee constitutes further evidence supporting the tachyonic interpretation, since as noted earlier, if the first knee is due to proton beta decay, we would expect to find a second knee due to alpha decay at an energy 50.9-times higher or $1.5 \times 10^{17}$ eV, which is very close to where it is found.

### 7. A n-p Decay Chain?

Explaining the ankle in terms of the hypothesis of a tachyonic electron neutrino obviously cannot be done in the same manner as the two knees, given that it is an upturn in $E^3 dF/dE$, not a downturn, so it could not be due to the onset of some new decay process. The explanation of the ankle starts with the observation that it is not possible to determine empirically whether the primary incident cosmic ray that initiates a shower of thousands of particles is a neutron or a proton. Furthermore, we observe that if ultra-high energy protons could beta decay, they would create neutrons having a slightly lower energy, which subsequently decay into protons, which decay back into neutrons, etc., creating a decay neutron–proton decay chain (NPDC). The NPDC would continue until the proton energy drops below the threshold for proton beta decay, i.e., the energy of the first knee.

Given a neutron lifetime of about 1000 seconds, one would normally not expect neutrons to reach us from very distant sources, but time dilation changes that expectation if the energy is high enough. Thus, at the energy $5 \times 10^{18}$ eV, a neutron would have its lifetime lengthened by roughly $5 \times 10^9$, yielding a mean free path before decay of roughly $mfp \sim 0.15 Mly$ for a single step in the decay chain. Given the ratio of the proton and electron masses, an energetic neutron might give up to the order of $1/2000$ of its energy in a single step of the decay chain, so there could easily be a thousand steps in the NPDC before the energy drops to that of the first knee. As a result, for the full NPDC, we might expect the particle to have an $mfp \sim 150 Mly$. That distance is comparable to the size of the Perseus–Pisces Supercluster, which stretches almost $300 Mly$. At one end of it is the Perseus Cluster (Abell 426), which is one of the most massive galaxy clusters within $500 Mly$ of us. We therefore expect that the higher the cosmic ray energy above $5 \times 10^{18}$ eV, the larger the mfp, so cosmic ray protons from more and more distant sources would reach us. This estimate would explain why the spectrum times $E^3$ abruptly turns up (softens) at this energy, creating the ankle. In addition, the idea of protons reaching us from more and more distant sources means that the spectrum would become more proton rich for energies well above the ankle. Evidence for the spectrum being proton rich at the highest energies is provided by the GZK cut-off occurring at the predicted energy for protons rather than that for heavier nuclei, which, as noted, would occur at an A-times higher energy. The preceding analysis corrects that in [57], when it was believed no GZK cut-off had been observed.

Most cosmic rays being charged particles have their directions randomized by the magnetic field of the galaxy, making it difficult to know the direction to their sources. With the hypothetical neutron–proton decay chain (NPDC), however, for which a cosmic ray particle spends most of its time as a neutron, its directionality would be mostly unaffected by the galactic magnetic field. As a result, we might suppose that cosmic rays that have energies close to the knee upon reaching Earth might, at least for not-too-distant sources, point back to their source. If one found such a source of cosmic rays having an energy at

the knee, its existence would offer evidence in support of the NPDC idea, and hence the tachyonic neutrino hypothesis.

Most cosmic ray physicists believe that no such cosmic ray sources have in fact been identified. While there were a dozen or so reports of Cygnus X-3 being such a cosmic ray source from the observations made in the 1970–1990 era, these are now considered to have been mirages in light of more recent negative results with more sensitive detectors. However, in [58], I explained why that mainstream view may be mistaken and that Cygnus X-3 really was and may still be a source of cosmic rays near the knee of the spectrum.

## 8. SN 1987A and Its Neutrinos

In this section, we describe what can be learned about the neutrino mass based on the data from supernova SN 1987A, the first visible one in or near our galaxy since the telescope was invented in 1608. About two to three hours before the visible light from this supernova reached us, a burst of neutrinos was observed at three neutrino observatories. The neutrinos arrived before the light because visible light is emitted from the supernova only after the shock wave reaches the stellar surface. We use the term neutrinos here to include antineutrinos as well, and in fact, the detectors observed many more $\bar{\nu}$ than $\nu$. The three detectors included Kamiokande II (12 events), IMB (8 events), and Baksan, (5 events), whose collective burst of 25 neutrinos lasted less than 13 s. Approximately five hours earlier, the Mont Blanc neutrino detector saw a five-event burst. Most physicists ignore that burst as being unrelated to SN 1987A, because of its early arrival time and its non-observation in the other detectors.

Shortly after the supernova was detected, it was recognized that the few dozen neutrinos observed could yield information about the neutrino mass, based on the measured energies and arrival times. The antineutrinos detected mostly were observed when they caused the reaction: $\bar{\nu}_e + p \rightarrow n + e^+$, with the $e^+$ detected based on the Cherenkov radiation it emits. The recorded quantities were the event time, approximate arrival direction, and the visible ($e^+$) energy, from which the neutrino energy can be found based on the previous reaction using: $E_\nu = E_{vis} + 1.3$ MeV. The usual analysis of these data yields only an upper limit on the electron neutrino mass, because it assumes the spread in neutrino arrival times reflects mainly a spread in their emission times from the supernova rather than a spread in their travel times, which would be negligible, if the three neutrino mass states were nearly degenerate, as normally assumed.

However, Huzita [64] and, a year later, Cowsik [65] made the opposite assumption, and they showed that if the emissions were near simultaneous from SN 1987A, the 25 observed neutrinos were all consistent with having one of two masses, which Cowsik cited as $m_1 = 4 \pm 1$ eV and $m_2 = 24 \pm 7$ eV. I rediscovered this result in a paper I wrote in 2012 [66], although my values were a bit different than Cowsik: $m_1 = 4.0 \pm 0.5$ eV and $m_2 = 21.4 \pm 1.2$ eV. The main reason for the difference between Cowsik's values and mine for $m_2$ was the result of my omission of the Baksan data in their computation on somewhat questionable grounds. This omission not only resulted in a smaller $m_2$, but an uncertainty that was probably unjustifiably small (1.2 eV versus his 7 eV). Recall that in the simultaneous emission analysis, the neutrino arrival times are assumed to be mainly a function of their travel time from the supernova, and the question of whether multiple masses are present is left for the data to decide, rather than imposing some favored model of the neutrino masses. It is true of course that current neutrino emission models from a core collapse supernova only have about 2/3 the neutrino emissions occurring in the first few seconds [67], but those models were developed after SN 1987A and need to be consistent with the standard view that emissions are spread over a $\sim 13$ s interval.

To see how the simultaneous emission analysis works, we define the arrival time $t$ relative to that of light, that is the neutrino travel time is: $t_{trav} = t + T$, where $T$ is the light travel time, approximately 168,000 years for SN 1987A. Since the detectors were not precisely synchronized, following the usual practice, we set $t = 0$ for the earliest-arriving neutrino in each detector. Under these assumptions, one can then compute individual

neutrino masses from relativistic kinematics based on their arrival time $t$ and energy $E$. We begin by observing that

$$v = \frac{L}{T+t} \approx \frac{L}{T}(1 - t/T) = c(1 - t/T) \tag{7}$$

and further that

$$v = c\sqrt{1 - 1/\gamma^2} \approx c\left(1 - \frac{m^2 c^4}{2E^2}\right) \tag{8}$$

Equating the two right-hand sides yields

$$m^2 c^4 = \frac{2E^2 t}{T} \tag{9}$$

Equation (9) means that in a plot of $1/E^2$ versus $t$ neutrinos of a given mass, $mc^2$ will cluster about a straight line through the origin of slope $M = 2/Tm^2c^4$. The mass for each cluster can then be found from the slope of the best-fit straight line—see Figure 3. Note that the fact that the two straight lines passing through the origin give good fits to the data verifies that the first arriving neutrino in each detector did have speeds very close to light, as was assumed. If there were any neutrinos having a specific value of $m^2 < 0$, they would cluster about a straight line through the origin with a negative slope, which in fact is the case for the five Mont Blanc neutrinos, as will be discussed later. It is difficult to assign a probability of finding such a result of two distinct masses by chance, given the small amount of data. However, it is noteworthy that Loredo and Lamb, who performed a Bayesian analysis of these data without any suggestion of two distinct masses, reported that two-component models for the neutrino signal are 100-times more probable than single-component models [68].

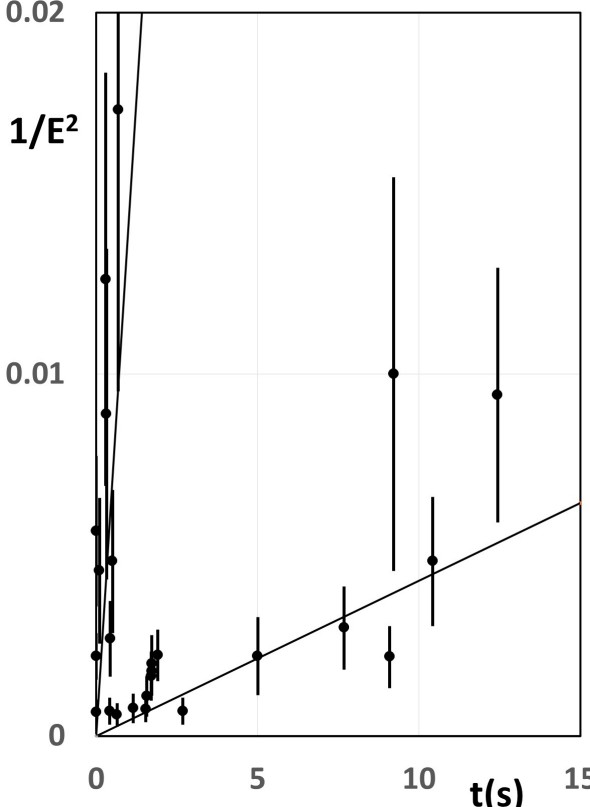

**Figure 3.** Plot of $1/E^2$ versus neutrino arrival time t for 25 neutrinos from SN 1987A providing evidence for two masses, based on the clustering of neutrino events near two straight lines.

### 9. The 3 + 3 Neutrino Model

Our main focus here will be on a model based on the SN 1987A neutrino data and the two neutrino masses as discussed in the previous section. Those large widely separated masses are obviously in direct conflict with the standard normal or inverted neutrino mass hierarchy. In both hierarchies, the $m^2$ values, while unknown, have values that are separated by no more than $dm^2 = 0.0024$ eV$^2$, that is the sum of the atmospheric and solar $dm^2$ values found from neutrino oscillation data. The primary basis for this standard neutrino mass hierarchy is that there is observed to be no third $dm^2$ value. In other words, in both the normal and inverted hierarchy, the oscillations corresponding to $dm^2 = dm^2_{atm} + dm^2_{sol}$ are experimentally indistinguishable from those for $dm^2 = dm^2_{atm}$. Continued adherence to the conventional hierarchies, however, ignores evidence that has been accumulating for a third oscillation frequency seen in some short baseline (sbl) experiments corresponding to $dm^2_{sbl} \sim 1$ eV$^2$ [69,70]. Moreover, the usual 3 + 1 way of incorporating a fourth sterile neutrino still does not give a satisfactory global fit in light of a tension between appearance and disappearance data [71], and the conflict remains even with two or three sterile neutrinos [72].

The preceding considerations led me to propose a 3 + 3 model [73], depicted in Figure 4, which was derived based on SN 1987A data. The model allows for having three very unequal neutrino masses, since it assumes the atmospheric and solar oscillations are within the two $m^2 > 0$ active–sterile doublets. The basis of the third doublet being tachyonic was so that the effective mass of the electron neutrino defined by $m^2_\nu = \Sigma|U_{ij}|^2 m^2_j$, could have the small negative value inferred from the cosmic ray data, as discussed earlier. The specific tachyonic mass value indicated in Figure 4 was based on a remarkable numerical coincidence, namely that the fractional separation of the two $m^2 > 0$ doublets turns out to be identical, namely $dm^2_1/m^2_1 = dm^2_2/m^2_2 = 5.0 \times 10^{-6}$ to within 4%. Therefore, it is reasonable to suppose the same value holds for the third $m^2 < 0$ doublet. Given the increasingly clear evidence for a third oscillation frequency corresponding to a $dm^2_{sbl} \sim 1$ eV$^2$, the value of the tachyonic mass would then be $m^2_3 = dm^2_3/5 \times 10^{-6} = -450^2$ eV$^2 \sim -0.2$ keV$^2$, to within a factor of two, given the uncertainty in $dm^2_{sbl}$. How such large masses can be accommodated with existing upper limits from neutrino mass experiments and cosmology are discussed in later sections.

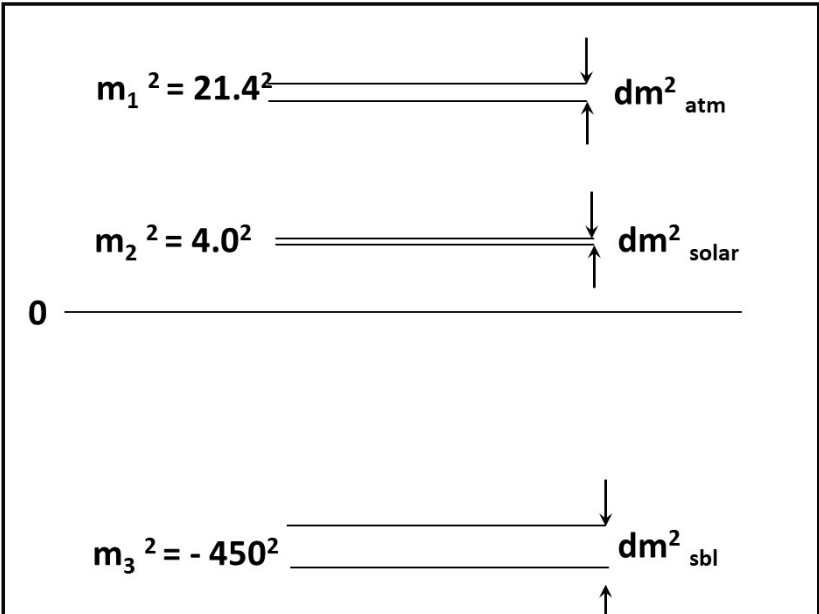

**Figure 4. The 3 + 3 model.** $m^2$ values in eV$^2$ for the three active–sterile doublets in the model and their splittings, $dm^2$. The choice of $m_3$ was made so that the three doublets have a common fractional splitting, $dm^2/m^2$. The plot is not to scale.

## 10. The $3 + 3 + LCR$ Model

Aside from the $3 + 3$ model just discussed, there is a second, not previously identified, model involving three active–sterile neutrino pairs, which includes tachyons. Alan Chodos suggested that neutrinos should satisfy a new symmetry principle, which he calls light cone reflection (LCR). This principle would require that neutrinos come in tachyonic and bradyonic pairs having the same magnitude of their mass $m = \sqrt{|m^2|}$ [74]. In that case, if the neutrino masses are also required to satisfy the three $dm^2$ observed in oscillation data, it follows that the $m^2$-level structure must then be as illustrated in Figure 5. Flavor state (effective) masses would of course be zero if, by symmetry, the $\pm m^2$ states were weighted equally, but given the size of the largest magnitude mass implied by this model, i.e., $m^2 = \pm \frac{1}{2} dm^2_{sbl} \sim \pm 0.5 \text{ eV}^2$, its presence might nevertheless be experimentally detectable in a direct mass experiment such as KATRIN, provided that the contributions to the effective mass from this pair of mass eigenstates are not negligible.

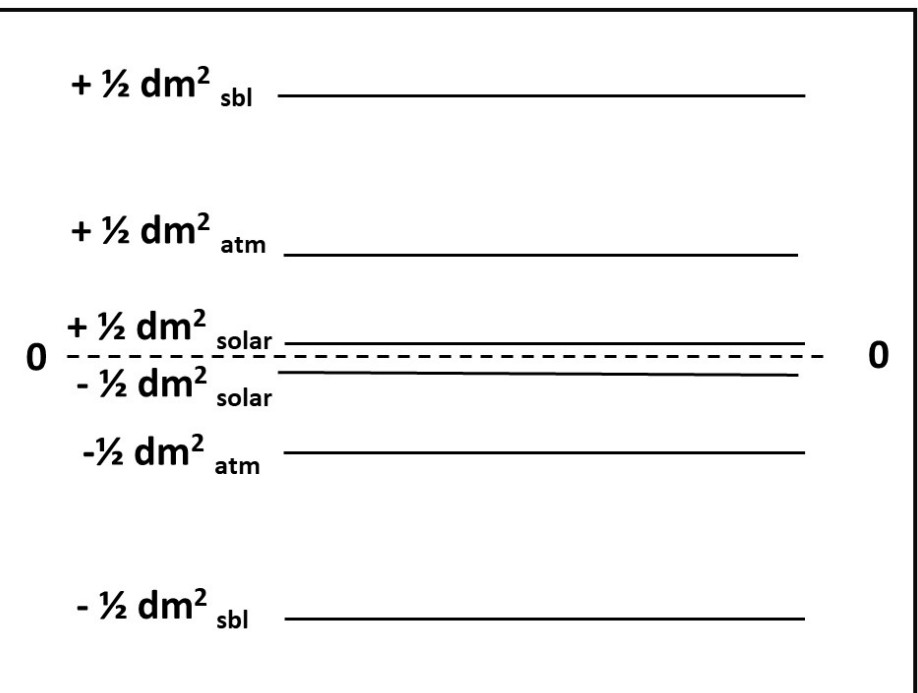

**Figure 5. The 3 + 3 + $LCR$ model.** Three active–sterile pairs having the indicated $m^2$ values under the assumptions of LCR symmetry and the $dm^2$ values observed in neutrino oscillation data (including $dm^2_{sbl}$). It is further assumed that oscillations are predominantly between states having the same $|m|$. Whether the active or sterile states have $m^2 > 0$ is unspecified. The plot is not to scale.

## 11. Evidence for the $3 + 3$ Model

Much of the rest of this paper consists of a description of evidence for the $3 + 3$ model. The extensive attention given to this evidence is justified in light of: (a) its magnitude and variety, (b) the lack of irrefutable conflicts with observations, (c) the new information since a previous summary of the evidence in [75], (d) the absence of any other tachyonic neutrino model that makes specific predictions aside from $3 + 3 + LCR$, and (e) the paucity of any other predictions of specific tachyon properties that are even testable.

### 11.1. The $m^2 > 0$ Masses in the $3 + 3$ Model

A confirmation of the two $m^2 > 0$ masses in the model comes from fits involving astrophysical data. Sterile neutrinos have long been considered a plausible possibility for dark matter, which holds galaxies and their clusters together. In 2014, Man Ho Chan and I showed, using the dark matter radial distribution for the Milky Way, which was inferred from the radial velocity curve, could be fit very well using a nearly degenerate gas

of neutrinos having a mass close to 21.4 eV [76]. We further showed that for four clusters of galaxies, the dark matter radial distributions holding them together could be fit using neutrinos having a mass close to 4.0 eV, these being the two $m^2 > 0$ masses in the $3 + 3$ model [76].

Additional evidence for the existence of one of the $m^2 > 0$ neutrino masses comes from work by Mohapatra and Sciama. In a 1998 unpublished manuscript [77], these physicists build on earlier work by Sciama [78]. They noted that the interstellar medium of the Milky Way galaxy is known to contain ionized hydrogen gas with properties that seem to defy common astrophysical explanations, and that the same is true for other galaxies. They therefore proposed that relic neutrinos, which are supposed to pervade the cosmic background, could explain the observed diffuse ionization if they had a mass of 27.4 eV and decayed into a lighter neutrino ($\nu_L$) plus a photon: $\nu \rightarrow \nu_L + \gamma$ with a lifetime $10^{22}$ s. In that case, the photons would have an energy of 13.7 eV and could provide enough ionizing photons to produce the observed diffuse ionization.

Since 1998, the puzzle of the observed diffuse ionization in the galaxy remains unexplained by conventional mechanisms. Thus, for example, Phan, Morlino, and Gabici [79] noted that, while cosmic rays are usually assumed to be the main ionization agent for the interior of molecular clouds, where UV- and X-ray photons cannot penetrate, a calculation of the amount of ionization produced by cosmic rays is more than ten-times less than what is observed. Mohapatra and Sciama's inferred neutrino mass for the sterile neutrinos in the galaxy is, of course, consistent with the mass that Cowsik found ($m_2 = 24 \pm 7$ eV) from his analysis of the SN 1987A data, although it is slightly too high to match the mass I used in developing the $3 + 3$ model derived from those same data, owing to my too small error bar on $m_2 = 21.4 \pm 1.2$ eV. The support for the $3 + 3$ model provided by the unexplained diffuse ionization in the galaxy has not been noted previously.

One further place where evidence for the presence of the two $m^2 > 0$ masses in the $3 + 3$ model might be found would be in oscillation data, where one might look for oscillations corresponding to $dm^2 = 21.4^2 - 4.0^2 \sim 450$ eV$^2$. As is well known, neutrino oscillations corresponding to a particular $dm^2$ result in a variation in the probability of neutrino appearance or disappearance of the form:

$$P \sim \alpha \times sin^2\left[1.27dm^2(\text{eV}^2)L(\text{km})/E(\text{GeV})\right] \tag{10}$$

Thus, seeing oscillations having as high a frequency as implied by $dm^2 \sim 450$ eV$^2$ would require both a short baseline $L$ and a high energy $E$. As of 2022, only two experiments have so far been sensitive to this high-frequency range. The MINOS group in 2019 reported seeing no evidence for oscillations with $dm^2 < 1000$ eV$^2$ with an amplitude larger than $\alpha \sim 0.02$ [80]. However, that negative result was based on the $3 + 1$ model (a single sterile neutrino oscillation), and it very well might not have picked up evidence for a double oscillation, namely both $dm^2 \sim 1$ eV$^2$ and $dm^2 \sim 450$ eV$^2$. In any case, the MINOS data would not have revealed oscillations with amplitude $\alpha < 0.02$.

A second recent experiment sensitive to oscillations with a $dm^2 < 1000$ eV$^2$ was performed by the MicroBooNE group [81]. This experiment did observe a hint of oscillations corresponding to $dm^2 \sim 1.4$ eV$^2$ (2.8$\sigma$), but for $dm^2 \sim 450$ eV$^2$, the situation was less clear. They analyzed the data by different methods, and according to one method (the Pandora analysis with 0 protons), their data were consistent with oscillations anywhere in the range $10 < dm^2 < 1000$ eV$^2$. Clearly, more oscillation experiments need to explore the little-explored high-$dm^2$ realm. While $m^2 < 0$ neutrinos can participate in oscillations as well [82], searching for a $dm^2 \sim 200,000$ eV$^2$ due to the $m_3$ mass in the model is out of the question. Fortunately, there are other ways to look for tachyonic neutrinos.

*11.2. The $m^2 < 0$ Mass in the $3 + 3$ Model*

When the $3 + 3$ model was first proposed in 2013, I accepted the conventional view that the 5 h early Mont Blanc burst (five neutrinos in 7 s with average energy $E_{avg} = 8 \pm 0.5$ MeV) was not connected to SN 1987A, even though I had been looking for signs of the tachyonic

mass postulated in the model. I had even been aware that since the burst was 282 min = 16,900 s early, then the inferred tachyonic mass using Equation (9) ($m^2 = -0.38$ keV$^2$) was within a factor of two equal to that in the 3 + 3 model. Finally, I was also aware that there was an explanation why that early burst was not seen in the other three detectors, namely that the Mont Blanc detector had a much lower energy threshold, which allowed it to detect the neutrinos in this burst, which all had much lower energies than those in the main burst.

One other potential problem with the Mont Blanc neutrinos being superluminal, with an excess above light speed $\delta = (v - c)/c = 3.2 \times 10^{-9}$ is the VPE Cherenkov radiation postulated by Cohen and Glashow [40]. However, the threshold for the process would be $E = m_e/\sqrt{\delta} = 9$ GeV, so neutrinos of 8 MeV energy would be well below the threshold, and they could reach Earth. Despite all the above, I still initially rejected the idea that the Mont Blanc burst was due to tachyonic neutrinos, a claim initially made by Giani in 1997 [83]. My initial rejection of the Mont Blanc burst being the tachyons in the 3 + 3 model was based on the requirement from Equation (9) that the spread in their energies would need to be:

$$\frac{\Delta E}{E} = \frac{2\Delta t}{t} = 2(7\,\text{s})/16,900\,\text{s} \approx 0.02\% \qquad (11)$$

which essentially required that the Mont Blanc neutrinos constituted a line in the SN 1987A spectrum of energy 8 MeV, their average value. Incidentally, the five Mont Blanc neutrinos strangely all had energies consistent with their average value, which means they are at least consistent with being monoenergetic. Still, in 2013, despite all the above, I regarded the notion of an 8 MeV line in the SN 1987A spectrum as being "inconceivable" [73]. By 2018, however, I understood how such an "inconceivable" neutrino line could be created during the supernova core collapse, and I provided evidence for its existence [84].

In that 2018 paper, I explained how the $Z'$ particle (sometimes called X17) [85,86] of mass 16.8 MeV could produce monoenergetic neutrinos (and antineutrinos) if there existed hypothetical cold dark matter X-particles of mass $m_X \sim 8$ MeV, which annihilated in the stellar core just prior to core collapse. The $XX$ annihilation essentially opened a portal between the dark matter and standard model matter worlds, which occurred when the supernova core reached a sufficiently high temperature, i.e., the threshold of the "$Z'$-mediated reactions":

$$X + X \rightarrow Z' \rightarrow \nu + \bar{\nu} \qquad (12)$$

$$X + X \rightarrow Z' \rightarrow e^+ + e^- \qquad (13)$$

Note that if the dark matter $X$ particles are cold ($v \ll c$), the $Z'$ are nearly at rest, and the leptons in Equations (12) and (13) are essentially monoenergetic, having $E \approx 8$ MeV. Having proposed a model for producing an 8 MeV neutrino line, it was essential to find corroborating evidence for it. One supporting piece of evidence provided in my 2018 paper involved gamma-ray data from the galactic center, which was a plausible place to find dark matter $X$ particles in addition to stellar cores. If indeed $X$ particles of mass $m_X$ resided there, this would lead to a broad enhancement to the spectrum up to an energy $E_\gamma = m_X$ through the annihilation of $e^+$ created by the reaction in Equation (13). A fit to the gamma ray spectrum showed that, in fact, the data gave a best fit for $m_X = 10^{+5}_{-1.5}$ MeV, which is consistent with $X$ particles of mass $m_X \sim 8$ MeV being partly responsible for the galactic center $\gamma-$rays, which in turn supported the $Z'$-mediated reaction as the cause of an 8 MeV neutrino line.

Of course, the best way to confirm such an 8 MeV neutrino line would be direct evidence for it in the SN 1987A neutrino data themselves, in particular the data from the Kamiokande II detector, the largest of the four recording data on the day of the supernova. Fortunately, in publishing their data for the neutrinos observed on that day, the Kamiokande II Collaboration showed plots of data taken during eight 17 min-long intervals before, and after the 12-event burst. That data consisted of 997 events in the form of dot plots of

arrival time versus number of "hits" or photomultipliers struck during a short time interval, which was a measure of the neutrino energy. The Collaboration considered these data merely detector background since it showed no obvious signs of events above 20 hits at any time besides the main 12-event burst, which was the way supernova neutrinos could be distinguished from the background consisting of radioactivity and cosmic rays.

However, as I described in [84], these 997 events were consistent with an 8 MeV neutrino line sitting atop a background, which if true, would lend strong support to the Mont Blanc neutrinos being the tachyons in the $3 + 3$ model, since as already noted, that is possible only if they were monochromatic with energy 8 MeV. This claimed 8 MeV neutrino line was deduced by comparing the energy distribution of the events recorded on the day of the supernova with published background data from the Kamiokande detector recorded in the months before and after SN 1987 A [87]. Despite finding a very large excess of events above a background with a shape, width, and central value, all being consistent with an 8 MeV neutrino line, this claim was controversial (and certainly not endorsed by the Kamiokande II Collaboration), since the excess occurred very close to the peak of the background. In a subsequent publication, however, I presented various reasons to believe the claimed line was genuine despite its falling close to the peak of the background and explained how various objections to its existence could be rebutted [88].

## 12. Neutrino Mass Experiments

In a previous (2019) summary of evidence for the $3 + 3$ model in [75], I noted that it is supported by fits to the observed tritium beta spectra that had been reported in three pre-KATRIN direct neutrino mass experiments. That claim turns out to have been a mistake on my part since those experiments simply were not sensitive enough to test the model. Nevertheless, it is worthwhile to explain how that mistake occurred, since it has bearing on the KATRIN experiment. Most neutrino mass experiments are based on observing the spectrum of tritium beta decay very close to the endpoint energy and seeing what value of the electron neutrino mass gives a best fit. Apart from various corrections, the phase space factor or the square of the Kurie function can be used to describe the differential spectrum, assuming the three neutrinos have virtually indistinguishable masses, and $m_\nu$ is their effective mass.

$$K^2(E) = (E_0 - E)\sqrt{R((E_0 - E)^2 - m_\nu^2)} \tag{14}$$

Here, $R$ is the ramp function $R(x) = x$ for $x > 0$, and $R(x) = 0$ otherwise. However, when fitting the spectrum to multiple distinguishable masses as in the $3 + 3$ model, one cannot use a single effective mass, but instead must use:

$$K^2(E) = \sum |U_{ej}|^2 (E_0 - E)\sqrt{R((E_0 - E)^2 - m_j^2)} \tag{15}$$

which may be thought of a a weighted average of three spectra with weighting factors $|U_{ej}|^2$ for each of the three masses $m_1, m_2, m_3$. In [75], I noted that the results from three experiments showed a "kink" in their spectra at around 20 eV before the spectrum endpoint, and I was able to obtain a good fit to the $3 + 3$ model masses to those three experiments' spectra when the weights for $m_1$ and $m_2$ were both $\sim$0.5, with only a tiny weight for the tachyonic $m_3$, which kept the effective mass very close to zero. These fits with equal weights for $m_1$ and $m_2$ resulted in a maximum amount of "kinkiness" in the differential spectrum near 20 eV, and they fit the data from the three experiments quite well. Regrettably, I later learned that those spectral anomalies in the data or kinks near 20 eV could be explained in terms of molecular tritium final states [89], and they probably had nothing to do with my model. On the other hand, the anomalies never would have appeared in my fits had I simply chosen a smaller spectral contribution from the $m_2 = 21.4$ eV mass, so there was at least no conflict with the $3 + 3$ model and those pre-KATRIN experiments, which simply were not sensitive enough to test it.

### 13. The KATRIN Experiment

The KATRIN experiment has been taking data since 2019, and it has published two values for the electron neutrino effective mass, based on the total amount of data so far accumulated at the time. In 2019, it reported an effective neutrino mass square value of $m^2 = -1.0^{+0.9}_{-1.1}$ eV$^2$ (tachyonic by $1\sigma$) [90], and in 2022, it reported $m_v = 0.26 \pm 0.34$ eV$^2$, which was also given in terms of an upper limit (at 90% CL) of $m_v < 0.8$ eV (90% CL) [91]. Unlike most previous experiments, which recorded a differential spectrum, KATRIN records its data as an integral spectrum, meaning that it records the integrated number of counts above a series of energies at a number of non-equally spaced set points chosen for maximum sensitivity.

The signature of the presence of the two $m^2 > 0$ masses in the integral spectrum is not spectral kinks (which would be the case in the differential spectrum), but rather, slight bumps or excess numbers of counts at a distance before the spectral endpoint $E_0$ equal to the two masses $m_1$ and $m_2$. The height of each predicted bump depends on the weight of the spectral contributions from the two $m^2 > 0$ masses, and these predicted deviations are shown in Figure 6 for the $3 + 3$ model masses. The smallest deviation from the single mass spectrum occurs when the contribution from $m_1 = 4.0$ eV is 94%, and the deviation is then less than 0.8% at all energies. The $m^2 < 0$ mass in the model influences the spectrum more subtly, because of its very small contribution to the overall spectrum.

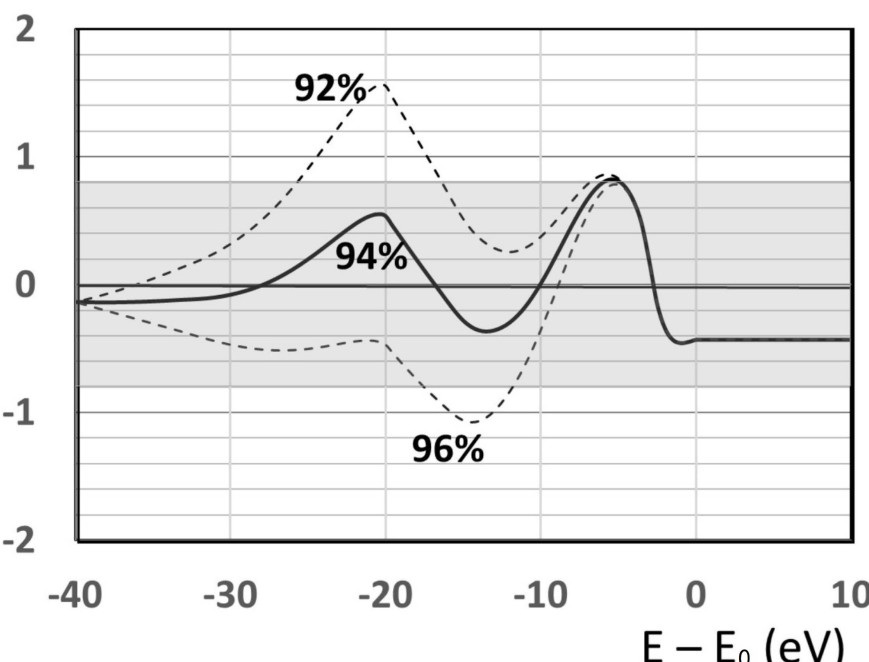

**Figure 6.** Predicted percentage deviations from the single mass $m = 0$ integrated spectrum for the $3 + 3$ model, for three choices of the contribution from the $m_1 = 4$ eV mass. The choice of 94% results in the smallest departure from the single mass spectrum, namely <0.8% at all energies.

In reporting its data KATRIN, shows both the actual spectrum, as well as the residuals (in standard deviations) to their best fit. Those residuals could reveal systematic departures from a single mass spectrum if they were large and not randomly distributed. In fact, the residuals in the KATRIN data from 2019 appear relatively random, but they also do give a strong hint of matching the predicted $3 + 3$ bumps, as I noted in a 2019 publication [75]. The agreement with the data KATRIN published in 2022 from their second campaign fails to show those same hints [88]. However, the alternate explanation of the data offered by

the $3 + 3$ model still remains viable, particularly in light of the very large number (37) of free parameters KATRIN used in their fit in [91].

That large number of free parameters occurs because KATRIN assumes different values of the signal, background, and endpoint energies for each of 12 detector rings and a single common neutrino mass, yielding $12 \times 3 + 1 = 37$ free parameters, unlike what was done in 2019 [90,91]. The use of so many free parameters by KATRIN could mask any $O(0.5\%)$ departures from the single mass spectrum that would occur if the $3 + 3$ model were a valid description of the data. That possible masking could occur not because KATRIN uses 37 parameters to fit their spectrum, which would be nonsensical. Rather, departures from the single mass spectrum would become less statistically significant when the spectral fits are performed for separate rings rather than all rings combined. The KATRIN data analysis team has expressed their commitment to performing their own fit to the $3 + 3$ model at some point during the course of the experiment, hopefully with a reduced number of free parameters. Alternatively, if there is no evidence for the $3 + 3$ model masses, KATRIN might alternatively be able to check whether the largest masses implied by the $3 + 3 + LCR$ model, i.e., $m^2 = \pm \frac{1}{2} dm^2_{sbl} \sim \pm 0.5$ eV$^2$, are present in the data, as well as whether there is any evidence for $m^2_{eff} = -0.11$ eV$^2$.

## 14. $3 + 3$ Model and Cosmology

The most challenging apparent conflict with the $3 + 3$ model masses is posed by cosmological constraints, whose consistency with these constraints is explained in much greater detail here than previously. As of 2022, the most constraining neutrino mass bounds yield: $\sum m_\nu < 0.09$ eV based on five different data sets [92], although one other 2022 assessment claims evidence for an actual value at a $4\sigma$ level: $\sum m_\nu = 0.23 \pm 0.06$ eV [93]. The basis for a constraint on $\sum m$ from cosmological observations comes from events occurring in the very young hot universe. At about one second after the big bang, neutrinos decouple and form the cosmic neutrino background (CNB). During that early hot ($T \sim$ MeV) epoch, neutrinos and antineutrinos would be produced in the decay of the $Z^0$ and $W^\pm$ bosons. In these decays, the three neutrino flavors would be produced in equal numbers, i.e., $n = n_e = n_\mu = n_\tau$, and the overall neutrino energy density $\rho$ and the number density $n$ in the early universe would be related through

$$\rho = n_e m_e + n_\mu m_\mu + n_\tau m_\tau = n \sum m_\nu \qquad (16)$$

where $\sum m_\nu$ is the sum of the three neutrino masses. However, notice that Equation (16) is the sum of the flavor (effective) masses, as Stecker [94] explicitly stated, not the sum of the mass eigenstate masses, as is almost always assumed. Of course, for the conventional neutrino picture where the three neutrino masses are nearly degenerate, this distinction makes very little difference, but this is not the case for the $3 + 3$ model, especially since one mass state is tachyonic.

Various authors, including Jentschura [95], Davies [96], and Schwartz [97], have considered tachyonic neutrinos as a candidate for dark energy, essentially a form of gravitational repulsion. The simplest way to understand why tachyonic neutrinos in the CNB might provide an effective gravitational repulsion is that the classical Newtonian formula for the force between any pair of neutrinos involves the product between two imaginary mass values, which is negative, meaning repulsion. Thus, if we have a mixture of tachyonic and bradyonic neutrinos making up the CNB, any concentrations of the latter promote the subsequent gravitational coalescence of matter to form structures, while the former impedes structure formation. Clearly, in $\sum m_\nu$ with both tachyonic and bradyonic flavors present, the former will appear with a negative mass, not an imaginary one, so as to allow for a partial cancellation of the two effects.

In order to understand what the sum over neutrino flavor state masses might predict for the $3 + 3$ model, we need to discuss the PMNS mixing matrix relating mass and flavor states. The PMNS matrix for the usual $3 + 0$ neutrino model is a $3 \times 3$ unitary matrix

characterized by three mixing angles and a CP-violating phase, while the $3 + 3$ model with its three active–sterile masses requires a $6 \times 6$ unitary matrix having 15 mixing angles and 10 phases. Given the difficulty of fitting 25 free parameters to obtain a good fit to all experiments, finding a $6 \times 6$ PMNS matrix required by the $3 + 3$ model has so far not been possible. While the exact form of this PMNS matrix connecting mass and flavor states remains to be described, we at least are assured from its unitarity that the sums of the mass-squared values for the mass and flavor states are identical, yielding:

$$m_e^2 + m_\mu^2 + m_\tau^2 = (4.0^2 + 21.4^2 - 200,000) \text{ eV}^2 \tag{17}$$

Finally, since the electron neutrino mass is so close to zero, Equation (17) leads to the $3 + 3$ model prediction that $m_\mu^2 + m_\tau^2 = -0.2 \text{ keV}^2$. This prediction of the model is many orders of magnitude from being testable, given the current upper limits on these two flavor masses [45]. Nevertheless, the main points of the preceding discussion are that given that the near-zero constraint of the sum of the neutrino masses from cosmology is a sum over flavor states and that tachyon masses enter the sum with a negative value, so the apparent conflict of the $3 + 3$ model with the cosmological constraint on $\sum m_\nu$ can be resolved provided that the tau and muon neutrino effective masses have the preceding stipulated relationship.

There is one other cosmological constraint on the relic neutrinos in the CNB, namely the effective number of neutrino species, which would include sterile neutrinos. Data from the Planck spacecraft has published the tightest bound to date on the this quantity: $N_{eff} = 3.15 \pm 0.23$ [98], which seems to be in conflict with a model having three light sterile neutrinos. However, Tang showed that this contradiction can be resolved if the sterile neutrinos are self-interacting, in which case, they can actually lower, not raise the effective number of neutrinos [99]. That is because as the Universe cools down, flavor equilibrium between active and sterile species can be reached after the Big Bang nucleosynthesis (BBN) epoch, but it causes a decrease of $N_{eff}$. In fact, Tang noted that based on his analysis, at least three eV-scale sterile neutrino species are needed to be consistent with the cosmological data then available (in 2015), which would offer further support for the $3 + 3$ model.

### 15. Summary and Future Tests

This article described the searches for tachyons, particles having $v > c$ and $m^2 < 0$. Initial searches were made for new particles having one of those properties, and apart from some false or ambiguous initial claims, they have yielded negative results. Following the pioneering work by Chodos et al. [32], most of the attention shifted to neutrinos as possible tachyon candidates. Here, the negative results of searches could better be described as inconclusive, rather than negative, at least as long as no neutrinos having either $m^2 > 0$ or $v < c$ are found. Even if some neutrino flavor is found to have $m^2 > 0$ or $v < c$, there would remain the possibility that one of the other flavors is a tachyon. Much of the second half of this review paper described the evidence for neutrinos as tachyons, and especially for the $3 + 3$ model, which I proposed in 2013 [73]. This model postulated three specific neutrino masses, one of which is a tachyon. These masses, are, of course, in conflict with the conventional neutrino paradigm, which requires three nearly degenerate masses, but nevertheless, a variety of observations have been cited in support of the three masses in the model.

Apart from the $3 + 3$ model, a separate earlier prediction was discussed for a specific electron neutrino effective mass from cosmic ray and other data [59]. Both predictions (and that of equal magnitude $\pm m_\nu^2$ from Chodos' proposed LCR symmetry) are so far consistent with the KATRIN data [75,88]. If that experiment by its conclusion fails to prove any of these predictions right or wrong, a new generation of direct mass experiments including Project 8 may be able to reach the necessary sensitivity [100]. Project 8 uses atomic, not molecular, tritium, and it has the potential to surmount the systematic limitations of current-generation methods, because atomic tritium avoids an irreducible systematic uncertainty

associated with the final states populated by the decay of molecular tritium. Project 8 hopes to achieve a 0.040 eV, neutrino-mass sensitivity (about five-times better than KATRIN).

Another way of testing the $3 + 3$ model masses would be to look for evidence of the $dm^2 = m_2^2 - m_1^2 \sim 450^2$ eV$^2$, in future oscillation experiments having sufficiently high energy and/or short baselines. One final suggested way to test the $3 + 3$ model would be to look for neutrinos associated in time with gamma-ray bursts, as was done by Huang and Ma in connection with the IceCube data [50,51]. Recall that they interpreted the apparent early arrival of some neutrinos relative to that of recorded gamma ray bursts, not as evidence for some neutrinos being tachyons, but rather their satisfying a Lorentz-violating (LV) dispersion relation between velocity and energy, which Huang and Ma assumed to be:

$$v(E) = c\left[1 - s_n \frac{n+1}{2}(E/E_{LV,n})^n\right] \tag{18}$$

Here, $n = 1, 2$ corresponds to the linear or quadratic dependence of the velocity on energy, $s_n = \pm 1$ is the sign factor of LV correction (+1 for superluminal and −1 for subluminal neutrinos), and $E_{LV,n}$ is the nth-order LV energy scale to be determined by fits to the data. Instead of the above interpretation of the IceCube neutrino data, if the GRBs come from sources having a known distance, $L$, one could test the $3 + 3$ model seeing if the data satisfy Equation (9), given the neutrino measured energies and the source distance for each of the three $3 + 3$ model masses. Specifically, on a plot of $1/E^2$ versus $t = L/c - t_{trav}$, one would look for data points falling near the three lines of the slopes corresponding to the three neutrino masses in the model, similar to the analysis performed for SN 1987A [66].

Aside from neutrinos associated with gamma-ray bursts, it might seem promising to use this same search method with the few dozen extragalactic supernovae that have been recorded since SN 1987A, but the number of neutrinos expected to reach Earth from them would be miniscule. Thus, for example, the nearest of these occurring in 1994 was 64-times further away than SN 1987A, and given the inverse square law, even a detector the size of Super-Kamiokande (25-times bigger than its descendant that observed 12 $\bar{\nu}$ from SN 1987A) would observe only $\sim 25 \times 12/64^2 = 0.075$ neutrinos from SN 1994. Of course, a new supernova in our galaxy would be quite another matter. The current generation of neutrino detectors is capable of detecting perhaps 10,000 neutrinos for a supernova at the galactic center in contrast to the mere few dozen observed for SN 1987A. Moreover, the next generation of detectors will increase that yield by another order of magnitude. Regrettably, given that only about three supernovae occur per century in the galaxy, the chances of this 84-year-old physicist witnessing such a marvelous event are not great. Nevertheless, the neutrino data from that next supernova will leave no doubt as to the truth or falsity of the $3 + 3$ model with its three specific masses inferred from SN 1987A, including the one with $m^2 < 0$.

A more extensive account of the "hunt for the tachyon" discussed in this review can be found in my recent popular-level book [101].

**Funding:** This research received no external funding.

**Institutional Review Board Statement:** Not applicable.

**Informed Consent Statement:** Not applicable.

**Acknowledgments:** The author wishes to thank Alan Chodos and Michael Kreisler for helpful comments on this article.

**Conflicts of Interest:** The author declares no conflict of interest.

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
