# Peer review of "A Review of Searches for Evidence of Tachyons"

_symmetry, doi:10.3390/sym14061198_

Round 1

Reviewer 1 Report

The article described the searches for the tachyons, specially the author described various experimental evidences for tachyons as neutrinos. I have the following issues:

(1) I have not found any new result in this manuscript. In particular, I request the author to clearly quote the new results of this work (if any), apart from bunching the previous results.

(2) If tachyons exist, then what about the Lorentz symmetry ? In particular, the Lorentz transformation between the lab frame and the tachyon particle does not exist anymore, as the tachyon has speed larger than light. Then how can one formulate the subject "Special theory of Relativity" when tachyons exist. I request the author to give some qualitative discussions (or quantitative) about this point.

   Based on these points, I recommend major revision. Thank you.

Author Response

I have now identified new results, which I flag in the paper (see attached).  These include:

  1. A new analysis in section VI of the consistency of cosmic ray data with the GZK cut-off, since the prior (1999) analysis assumed the cut-off was evaded
  2. A new point of support for the 3 + 3 model provided by diffuse ionization in the galaxy in section XI A.
  3. A new “3+3+LCR” model discussed in section X.
  4. A new detailed explanation of the comparison of the 3 + 3 model with cosmological constraints in section XIV with respect to both the sum of the neutrino masses and the effective number of neutrinos

On the reviewers second point I have added some discussion to the second paragraph of section V to clarify this matter.

Reviewer 2 Report

The paper devoted to the review of ideas and experiments aimed to search for empirical evidences for the existence of tachyons. In fact, the manuscript consists of two parts: the first one is the review of some earlier experiments, and the second one describes the properties and possible empiric features of the so-called 3+3 model proposed by the Author. The paper is well written, a wide variety of experiments and their conceptual interpretations is considered. I think this paper can be published in its present form.

Author Response

Thank you for approving the paper as is.  Various improvements have nevertheless been made.

Reviewer 3 Report

A very interesting review of searches for hypothetical tachyons - faster than light particles. The topic may be of definite interest for Symmetry readers and the review is written in a very clear and comprehensive way. It makes possible to recommend to accept the paper in the present form. The only comment, which goes beyond the scope this paper and should be considered as a question for future thinking, is the possibility to remove tachyon solutions in Quantum Field Theory by proper choice  vacuum.

Author Response

(The authors gave the same response as above.)

Round 2

Reviewer 1 Report

(1) In regard to my first query of the previous report, the author has given satisfactory answer. 

(2) However in regard to my second query, i.e how do one formulates the subject "Special Theory of Relativity" in presence of tachyons,  -- the author has NOT given any proper answer. The author has written - "it is not possible to do a Lorentz Transformation (LT) from the lab frame to the tachyon rest frame. One can, however, do a LT of the tachyon neutrino energy in going from the lab frame (E) to the rapidly moving proton rest frame". I do not understand this statement. 

My particular questions are -- (a) In presence of tachyons, the Lorentz factor \sqrt{1-v^2/c^2} (between lab frame and tachyon frame) becomes imaginary, as v>c for tachyons. How do we resolve this problem ? (b) The sqaured mass of tachyons are negative. Then how do we formulate the famous equation E = mc^2 ? 

I request the authors to properly answer these two questions. After that, I can recommend the paper for publication, otherwise not. 

Author Response

In response to his point #(2)

A transformation to the tachyon rest frame from the lab frame would require a v>c so we can never transform to a frame where a tachyon is at rest, any more than we could transform to a frame where a photon is at rest.  A tachyon (or a photon) has no rest frame.

In response to his (a) stated below

"(a) In presence of tachyons, the Lorentz factor \sqrt{1-v^2/c^2} (between lab frame and tachyon frame) becomes imaginary, as v>c for tachyons. How do we resolve this problem ?"

As I just said above we cannot (and need not) transform to the tachyon rest frame (or a photon's rest frame), because it doesn't have any, so I am unclear what problem the reviewer is trying to solve.  Were the reviewer to apply his question in (a) to a photon he would apparently rule out their existence since you get infinity when transforming to their rest frame.

Finally in response to his: (b) The sqaured mass of tachyons are negative. Then how do we formulate the famous equation E = mc^2 ? 

I am unclear what the reviewer is asking here.  For a tachyon in the lab it has v > c which yields an imaginary gamma factor and an imaginary m, yielding a real energy based on E=mc^2.  Exactly what is the problem?

I request the authors to properly answer these two questions. After that, I can recommend the paper for publication, otherwise not. 

Round 3

Reviewer 1 Report

The author has tried to answer the queries. However I am still confused by the following question -

(1) Does not the subject "Special Theory of Relativity" face some problem due to the existence of Tachyon which moves faster than light ?

However, provided that the authors tried to answer the query and the other referees recommended for publication, I will not hold the publication of this review article further. So I recommend accept.